# SPAZER: Spatial-Semantic Progressive Reasoning Agent for Zero-shot 3D Visual Grounding

**Zhao Jin**[1]  **Rong-Cheng Tu**[1,§]  **Jingyi Liao**[1]  **Wenhao Sun**[1]
**Xiao Luo**[2]  **Shunyu Liu**[1]  **Dacheng Tao**[1,§]

[1] College of Computing and Data Science, Nanyang Technological University, Singapore
[2] University of California, Los Angeles

{zhao.jin, rongcheng.tu, jingyi012, wenhao006}@ntu.edu.sg
xiaoluo@cs.ucla.edu, {shunyu.liu, dacheng.tao}@ntu.edu.sg

## Abstract

3D Visual Grounding (3DVG) aims to localize target objects within a 3D scene based on natural language queries. To alleviate the reliance on costly 3D training data, recent studies have explored zero-shot 3DVG by leveraging the extensive knowledge and powerful reasoning capabilities of pre-trained LLMs and VLMs. However, existing paradigms tend to emphasize either spatial (3D-based) or semantic (2D-based) understanding, limiting their effectiveness in complex real-world applications. In this work, we introduce SPAZER — a VLM-driven agent that combines both modalities in a progressive reasoning framework. It first holistically analyzes the scene and produces a 3D rendering from the optimal viewpoint. Based on this, anchor-guided candidate screening is conducted to perform a coarse-level localization of potential objects. Furthermore, leveraging retrieved relevant 2D camera images, 3D-2D joint decision-making is efficiently performed to determine the best-matching object. By bridging spatial and semantic reasoning neural streams, SPAZER achieves robust zero-shot grounding without training on 3D-labeled data. Extensive experiments on ScanRefer and Nr3D benchmarks demonstrate that SPAZER significantly outperforms previous state-of-the-art zero-shot methods, achieving notable gains of **9.0%** and **10.9%** in accuracy. Our codes are available at https://github.com/JZ-9962/SPAZER.

## 1 Introduction

3D Visual Grounding (3DVG) focuses on accurately localizing the referred object in the 3D scene based on a user-provided query text. This task demands a comprehensive understanding of both natural language and spatial structure of the 3D scene, serving a pivotal role in various real-world applications, such as embodied robotics [23, 24] and augmented reality [6, 22]. In the early phase, 3DVG approaches [6, 57, 53, 25, 16, 1] typically follow a fully supervised learning paradigm, training models using manually annotated textual descriptions and 3D bounding boxes. Despite their effectiveness, the reliance on large-scale labeled 3D data, which is expensive and labor-intensive to obtain, significantly limits their scalability and practical adoption.

To reduce the dependence on costly 3D training data, zero-shot 3D visual grounding has been actively explored, largely facilitated by recent advances in Large Language Models (LLMs) and Vision-Language Models (VLMs). By leveraging the general world knowledge and reasoning

---

§ Co-corresponding authors

39th Conference on Neural Information Processing Systems (NeurIPS 2025).

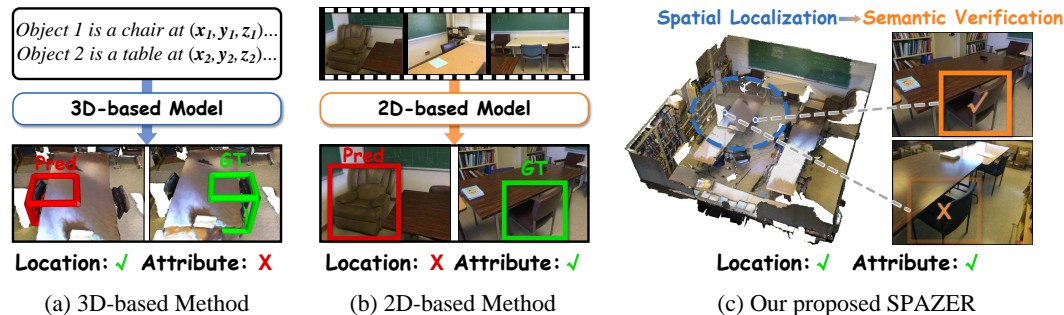

Figure 1: Schematic comparison between our proposed SPAZER and existing zero-shot 3DVG approaches. Compared to (a) 3D-based methods [49, 54, 22] that focus on spatial localization but tend to ignore fine-grained object attributes, and (b) 2D-based methods [48] that match visual appearance yet lack precise spatial awareness, (c) our approach progressively performs 3D spatial localization and 2D semantic verification, enabling more reliable grounding results.

abilities of pre-trained LLMs or VLMs, these methods achieve zero-shot 3DVG without 3D-specific training. Since native 3D data (*e.g.,* point clouds) cannot be directly interpreted by LLMs or VLMs, a key challenge in this setting is bridging the modality gap—*i.e.*, transforming 3D data into proper representations that language or image models can understand. To address this, existing methods mainly follow two design paradigms: **3D-based** and **2D-based**. 3D-based methods [49, 54, 22] generally operate on point clouds by extracting object-level textual descriptions with categories and 3D positions, which are then processed by LLMs for reasoning. In contrast, 2D-based methods [48] leverage 2D video sequences (which are inherently part of the scanned 3D dataset) to understand 3D scenes, using pretrained VLMs to match visual frames with the language query.

Although these two categories of zero-shot methods have shown promising results, they generally specialize in either **spatial** reasoning or **semantic** understanding, limiting their effectiveness in complex 3D scenes that require both. As illustrated in Fig. 1, the query text not only indicates the spatial location of the target chair — "*under the first dark brown table*" — but also highlights its visual attribute as "*brown leather*". Since 3D-based methods (a) typically utilize 3D coordinate-based textual description, they focus more on spatial reasoning yet struggle to distinguish objects with similar location but differing appearance. Conversely, 2D-based methods (b) tend to match visually relevant objects frame-by-frame, but lack explicit awareness of object positions within the 3D scene. However, accurate 3D visual grounding requires the effective coordination of both spatial and semantic reasoning, especially in dynamic real-world environments.

To address the above issue, we propose **SPAZER**, a **S**patial-Semantic **P**rogressive reasoning **A**gent for **ZE**ro-shot 3D visual g**R**ounding. Inspired by cognitive neuroscience findings [4, 50] that spatial reasoning and object recognition follow distinct neural streams in the human brain, our agent SPAZER decouples spatial and semantic reasoning based on distinct modalities. In the spatial reasoning stage, we propose a holistic multi-view rendering-based 3D representation. This enables our agent to directly observe and understand the 3D scene itself instead of processed textual descriptions. Leveraging this representation, SPAZER first identifies the optimal viewpoint for observing the target object, then determines potential candidate objects that match the query description from the selected perspective. Meanwhile, our agent incorporates a retrieval-augmented anchor filtering and annotation strategy to provide reliable guidance for spatial reasoning. After coarsely identifying possible candidates in the scene, relevant 2D close-up images will be retrieved from raw scanning data for fine-grained semantic reasoning. Following 3D spatial reasoning to identify candidate object locations and 2D semantic reasoning for detail verification, SPAZER realizes 3D-2D joint decision-making to produce the final grounding result. In summary, our contributions are as follows:

- We introduce SPAZER, a training-free 3D visual grounding agent that integrates 3D spatial localization and 2D semantic verification through a coarse-to-fine progressive reasoning process, enabling efficient 3D-2D joint decision-making.

- We propose to represent 3D point clouds in holistic rendered views instead of traditional object-level descriptions to enable VLMs direct observe and understand 3D scenes.

- We propose a retrieval-augmented candidate object screening strategy to improve the spatial reasoning ability of our agent, which exhibits enhanced robustness in object localization.

- Extensive experiments demonstrate the superior performance of our method, largely surpassing previous state-of-the-art (SOTA) by **9%** on ScanRefer and **10.9%** on Nr3D. SPAZER is even comparable to SOTA supervised methods on Nr3D, with a marginal gap ($\sim 1\%$) in overall accuracy.

## 2 Related work

**Supervised 3D visual grounding.** As a fundamental 3D vision-language task, 3D Visual Grounding (3DVG) has gained continuous research attention in recent years [6, 1, 57, 53, 3, 16, 47, 8, 56, 15, 30, 11, 44, 51, 14, 55, 39]. The core problem in 3DVG lies in accurately localizing target object in the 3D scene based on the given referring description. Traditional methods typically tackle this task under a supervised learning paradigm, mainly following two-stage or single-stage network architectural designs. *Two-stage* methods [6, 57, 53, 7, 8] adopt a detect-and-match strategy: they first extract object-level features from the scene using pre-trained 3D detection or segmentation models [32, 20, 34], and then fuse these with language features to compute the best-matching object. In contrast, *single-stage* methods [25, 16, 47, 43] directly fuse point cloud and textual features to predict the target object's 3D bounding box in an end-to-end manner. In recent years, with the development of vision-language pre-training (VLP) technologies, an increasing number of methods [21, 59, 13, 33, 58, 41] have adopted cross-modal joint pre-training for the 3DVG task. As the pre-training data continues scale up, these models demonstrate increasingly stronger generalization capabilities across datasets.

**LLM-based agent for zero-shot 3DVG.** Although supervised 3DVG methods exhibit encouraging performance, their success remains highly dependent on the availability of large-scale high-quality annotations, which remain relatively limited in current 3D datasets. Inspired by the success of LLM agents in various multimodal reasoning and generation tasks [26, 46, 35, 37, 38, 36], zero-shot 3DVG has emerged as a promising alternative to alleviate the reliance on 3D training data. Early approaches build agent workflows based on LLMs (*e.g.,* ChatGPT [29]), decomposing the 3DVG task into a series of sub-tasks that can be handled by the LLM. For instance, LLM-Grounder [49] formulates zero-shot 3DVG as a tool-augmented reasoning task. It leverages an LLM to decompose complex queries, interact with 3D perception tools (e.g., OpenScene [31]), and perform spatial reasoning to identify the referred object. Building on the concept of visual programming [12], ZSVG3D [54] prompts LLMs to generate interpretable visual programs composed of modular spatial functions, enabling enhanced structured reasoning. EaSe [27] employs LLMs to generate symbolic spatial relation encoders in the form of executable Python code. While LLM-based 3DVG agents are effective at modeling spatial relations, their performance is constrained by limited visual understanding capabilities, which are also crucial for accurately grounding objects in complex 3D scenes.

**VLM-based agent for zero-shot 3DVG.** To complement the limitations of LLM-based agents in visual perception, recent studies have turned to VLM-based agents to enhance semantic understanding in 3D scenes. SeeGround [22] proposes a hybrid 3D representation composed of spatially localized object descriptions and query-aligned local renderings, which provides additional visual contexts for the VLM, thereby enhancing grounding performance in complex 3D environments. Different from aforementioned methods that are build upon 3D point clouds, VLM-Grounder [48] proposes to realize 3DVG using only 2D video sequences, which are readily available in 3D scanned datasets. It stitches multiple frames to reduce input token length and combines various off-the-shelf tool models to obtain 3D bounding boxes of VLM-selected objects. While 2D camera images offer finer-grained semantic cues compared to rendered views, the limited 3D spatial layout and relation awareness makes VLM-Grounder prone to spatial reasoning errors. In summary, although promising zero-shot 3DVG results have been achieved, existing methods typically rely solely on either 3D geometry representation or 2D camera images, and the potential of their joint exploitation for more comprehensive spatial-semantic reasoning remains underexplored.

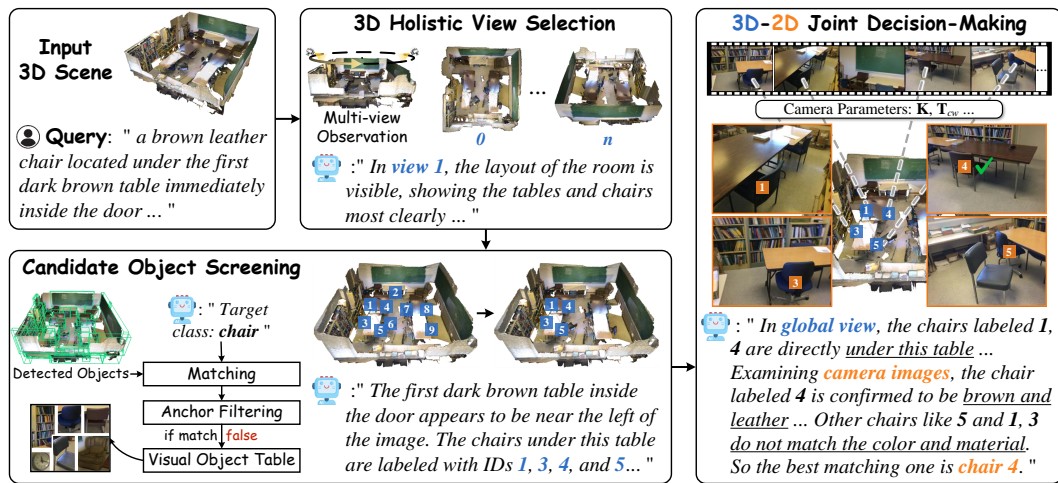

Figure 2: Overview of the SPAZER framework. Given input 3D scene and a query text of the target object, SPAZER performs zero-shot 3D visual grounding through a spatial-semantic progressive reasoning process. It first conducts 3D Holistic View Selection (Sec. 3.2) and Candidate Object Screening (Sec. 3.3) to identify spatially plausible candidates by globally analyzing the 3D scene. Then, 2D-3D Joint Decision-Making (Sec. 3.4) is performed by leveraging 2D camera views for fine-grained semantic verification of the candidates, ultimately grounding the target object without any 3DVG-specific training.

## 3 Method

### 3.1 Overview

In the 3D visual grounding task, given a query textual description, the objective is to localize the referring target object within the 3D scene. Our proposed agent SPAZER solve this task in a zero-shot paradigm, *i.e.*, directly performing inference and requiring no training on 3D visual grounding datasets. It integrates a VLM as the core for reasoning and decision-making, equipped with a suite of 3D and 2D perception tools that facilitate scene understanding at multiple levels of granularity. The complete architecture of SPAZER is illustrated in Fig. 2. It begins with a holistic analysis of the scene's global 3D geometry to determine potential candidates that align with the query text. This process is achieved through two components: 3D Holistic View Selection (Sec. 3.2) and Candidate Object Screening (Sec. 3.3). Subsequently, SPAZER further verifies the semantic details of candidate objects using 2D camera views, and derives the refined grounding result through 2D-3D Joint Decision-Making in Sec. 3.4.

### 3.2 3D holistic view selection

Since VLMs are not capable of directly interpreting 3D data, existing methods either convert 3D scenes into textual narratives [49, 54], or combine textual descriptions with locally rendered perspective views [22]. Although such text-centric 3D representations provide explicit object coordinates, their reliability is highly sensitive to the accuracy of predicted bounding boxes. Inaccuracies in object category or location predictions can severely compromise the spatial reasoning process. To overcome these limitations, we propose to directly empower the agent with multi-view observation capabilities, enabling it to understand the 3D scene itself without relying on intermediate textual representations.

**Multi-view observation.** In this step, SPAZER leverages 3D rendering tools to generate holistic views of the input point cloud from multiple perspectives. Firstly, the viewpoint is set directly above the center of the scene to obtain the Bird's-Eye View (BEV). To ensure that the BEV image fully captures the entire scene, the distance between the viewpoint and the scene center is calculated as

$$d = \frac{1}{2} \times \frac{\max(l_x, l_y)}{\tan(\theta/2)}, \tag{1}$$

where $l_x$ and $l_y$ denote the scene extents along the x-axis and y-axis, respectively, and $\theta$ represents the rendering camera's Field of View (FoV). Furthermore, we introduce $n$ additional viewpoints from different perspectives to observe the scene. The $n$ viewpoints are distributed uniformly along a circular path above the scene, each separated by $360°/n$, all sharing a fixed oblique viewing angle $\alpha = 45°$ and the distance to the scene center equals to $d$. For example, when $n = 4$, the viewpoints are placed at the front, back, left, and right of the scene, providing four angled top-down renderings.

**View selection.** After obtaining the all $(n + 1)$ rendered global views $\mathcal{I}_{3d} = \{I_{3d}^0, I_{3d}^1, \ldots, I_{3d}^n\}$, we feed them along with the query text $q$ into the VLM, which is prompted to select the optimal viewpoint that best captures the query-described object:

$$I_{3d}^* = \text{VLM}(\mathcal{I}_{3d}, q), \tag{2}$$

where $I_{3d}^*$ denotes the selected 3D rendering view. After this step, our agent has holistically observed the scene and rendered an optimal view, which serves as the global 3D representation for the subsequent progressive localization of the target object.

## 3.3 Candidate object screening

Leveraging the selected optimal viewpoint, SPAZER proceeds to infer the position of the target object that satisfies the query language. To facilitate this procedure, we follow the previous setting [54, 22] to utilize an off-the-shelf 3D network to extract object layout information, which can be represented as: $\mathcal{O} = \{(b_i, c_i) \mid i = 1, 2, \ldots, N\}$, where $b_i$ represents the 3D bounding box of the $i$-th object, and $c_i$ denotes its associated class label.

**Retrieval-augmented anchor filtering.** Since VLMs struggle to directly output precise spatial coordinates, we provide visual anchors as cues by leveraging object layout information. Firstly, the agent filters out irrelevant objects by analyzing the query text to identify the target object class $c^*$. Subsequently, a text-based matching algorithm is employed on $c^*$ and each detected object category $c_i$ to compute the similarity score $s(c^*, c_i)$. And the object class with the highest similarity score will be used to filter out irrelevant objects whose predicted class not equaling to it. However, due to the free-form nature of the query text, object categories may be semantically ambiguous (*e.g.*, a "cabinet" might be referred to as a "shelf" or "board" in the query). As a result, relying solely on text-based matching can lead to unreliable results. To address this issue, we propose a visual retrieval-augmented matching mechanism. We first set a threshold $\tau$ to filter out low-confidence matching results:

$$\hat{c} = \begin{cases} c_{i^*}, & \text{if } s(c^*, c_{i^*}) \geq \tau \\ \varnothing, & \text{otherwise} \end{cases} \quad \text{where} \quad i^* = \underset{i=1,\ldots,N}{\arg\max} \; s(c^*, c_i). \tag{3}$$

If a valid $\hat{c}$ is obtained, only the detected objects with predicted class label equaling to $\hat{c}$ will be retained, forming the subset $\mathcal{O}_{\text{cat}} = \{(b_i, c_i) \in \mathcal{O} \mid c_i = \hat{c}\}$. When $s(c^*, c_{i^*}) < \tau$, our agent will switch from text-based matching to vision-based matching to improve the reliability. It first constructs a visual object table, which stitches the cropped object images based on the position of each 3D bounding box. This step shares a similar process as camera view mapping in Sec. 3.4, which will be elaborated later. Then the VLM is prompted to automatically filter out objects that do not match the target category based on the visual object table. In this way, when unreliable object category predictions occur, our agent can still re-evaluate based on visual cues, thereby enhancing robustness.

**Candidate screening.** After filtering out irrelevant objects, the remaining object IDs will be annotated on the previously selected view images as visual anchors. For each ID, the 3D-to-2D coordinate transformation is computed based on the rendering parameters of the current view, and the annotation is placed at the center of the box. The VLM is then prompted to identify potential anchors from the annotated IDs. At this stage, although it is feasible to directly select the target object based on the query text, 3D renderings often lack fine-grained details and struggle to tell view-dependent spatial relationships. So we relax this process to a coarse selection of the Top-$k$ most likely objects:

$$\mathcal{O}_{\text{Top-}k} = \text{Top-}k\left(\text{VLM}(I_{3d}^*, \mathcal{O}, q)\right). \tag{4}$$

In subsequent experiments, we adjust the value of $k$ to ensure a balance between increasing the likelihood of covering the target object and avoiding excessive image inputs and token overhead.

## 3.4 3D-2D joint decision-making

In the preceding phase, SPAZER has conducted a initial reasoning and identified the coarse candidate pools. Next, we incorporate informative 2D camera views to enrich semantic context, thereby

facilitating 3D-2D joint decision-making. The 2D camera images are commonly available in original RGB-D scans, capturing detailed visual appearance of the scene. However, due to the large number of images (*e.g.*, up to thousands of images per scene in ScanNet [9]), how to efficiently sample keyframes remains a challenge in existing studies [48, 58, 33]. Benefiting from the global 3D spatial reasoning process, our agent has obtained $k$ candidate anchors. Accordingly, we efficiently sample $k$ corresponding keyframes based on the 3D-to-2D mapping of the anchor positions.

**Camera view mapping.** Our agent utilizes the camera intrinsics along with the extrinsics to compute the mapping from 3D points to 2D camera views. To determine the most informative view for each candidate anchor, we project its associated 3D bounding box $\mathbf{P}_{3D} \in \mathbb{R}^{3 \times 9}$ onto camera view images and evaluate per-view visibility. Specifically, we extract nine key points from each anchor's bounding box, including its eight vertices and the center. These 3D points, defined in world coordinates, are first converted to homogeneous coordinates $\widetilde{\mathbf{P}}_{3D} \in \mathbb{R}^{4 \times 9}$ (*i.e.,* add 1-padding for affine transformation). Then we perform a transformation analogous to point cloud registration [45, 18, 19] to convert them into the camera coordinate system using the extrinsic matrix $\mathbf{T}_{cw} \in \mathbb{R}^{4 \times 4}$, resulting in $\mathbf{P}_{\text{cam}} = \mathbf{T}_{cw} \cdot \widetilde{\mathbf{P}}_{3D}$. Next, we project the 3D points in camera coordinates onto the 2D image plane using the intrinsic matrix $\mathbf{K} \in \mathbb{R}^{3 \times 3}$. For each point, let $\mathbf{p}_{\text{cam}} = [x_c, y_c, z_c]^T$ be its 3D position in the camera frame. The corresponding 2D pixel coordinate $\mathbf{p}_{2D} = [u, v]^T$ is computed by:

$$\begin{bmatrix} u \\ v \\ 1 \end{bmatrix} = \mathbf{K} \cdot \begin{bmatrix} x_c/z_c \\ y_c/z_c \\ 1 \end{bmatrix}, \tag{5}$$

where $u$ and $v$ represent the horizontal and vertical coordinates of the projected point in the image plane. To handle occlusions, we compare the projected depth with the depth map. A point is considered visible if its projected coordinate $(u, v)$ lies within the image bounds and the depth difference is negligible. By computing the camera view mapping across multiple images, anchor-relevant views can be determined. Since an anchor object may appear in multiple views, we compare the number of visible projected points across all camera images to select the one that observes the most complete object bounding box as its corresponding 2D view. Finally, the selected 2D camera images for all candidate anchors are obtained as the set $\mathcal{I}_{2d} = \{I_{2d}^1, I_{2d}^2, \ldots, I_{2d}^k\}$, where each $I_{2d}^j$ corresponds to the most informative view associated with the $j$-th candidate anchor.

**Joint decision-making.** After obtaining corresponding camera view images of the selected Top-$k$ candidate anchors, our agent performs joint decision-making by leveraging both 3D global rendering and 2D camera images. To enable the VLM to seamlessly integrate different perceptual modalities during reasoning, the same candidate object is annotated with a consistent ID across rendered image and different video frames. In this process, SPAZER leverages the global 3D rendering for spatial reasoning, focusing mainly on re-evaluating the view-independent relations (*e.g.,* in the middle/corner of the room). Meanwhile, utilizing 2D camera views, it further verifies semantic details of candidate objects (*e.g.,* color, shape, material) and infers view-dependent relations (*e.g.,* left, right). After jointly considering both modalities for decision-making, the agent identifies the best-matching target and outputs its 3D bounding box:

$$b^* = \text{VLM}(I_{3d}^*, \mathcal{I}_{2d}, \mathcal{O}, q). \tag{6}$$

## 4 Experiment

### 4.1 Experimental setup

**Datasets.** We evaluate our method on two widely-used 3D visual grounding benchmarks: ScanRefer [6] and Nr3D [1], which are built upon the ScanNet [9] dataset. **ScanRefer** comprises 51,583 human-written natural language descriptions across 800 indoor scenes from ScanNet, each referring to a specific object in 3D space. Queries are classified into "Unique", where only a single object of the target class exists, or "Multiple", where other same-class distractors are present, requiring more precise discrimination. **Nr3D** contains 41,503 descriptions collected via a two-player reference game aimed at improving linguistic precision. Each query refers to a target object that is always accompanied by at least one distractor, and queries are labeled as either "Easy" (one distractor) or "Hard" (multiple distractors). Moreover, queries are also categorized as "View-Dependent" (Dep.) or "View-Independent" (Indep.) depending on whether understanding spatial relations like "left" or "right" requires specific camera viewpoints. To enable fair comparison and reduce expenditure, our main experiments are conducted on the same ScanRefer and Nr3D subsets as [48].

Table 1: Quantitative comparison with supervised and zero-shot 3DVG methods on **ScanRefer** [6].

| Method | Zero-shot | LLM/VLM | Unique | | Multiple | | Overall | |
|---|---|---|---|---|---|---|---|---|
| | | | Acc@0.25 | Acc@0.5 | Acc@0.25 | Acc@0.5 | Acc@0.25 | Acc@0.5 |
| ScanRefer [6] | ✗ | - | 67.6 | 46.2 | 32.1 | 21.3 | 39.0 | 26.1 |
| 3DVG-Transformer [57] | ✗ | - | 81.9 | 60.6 | 39.3 | 28.4 | 47.6 | 34.7 |
| BUTD-DETR [16] | ✗ | - | 84.2 | 66.3 | 46.6 | 35.1 | 52.2 | 39.8 |
| ChatScene [13] | ✗ | Vicuna-7B | 89.6 | 82.5 | 47.8 | 42.9 | 55.5 | 50.2 |
| Video-3D LLM [58] | ✗ | LLaVA-Video 7B | 88.0 | 78.3 | 50.9 | 45.3 | 58.1 | 51.7 |
| GPT4Scene [33] | ✗ | Qwen2-VL-7B | 90.3 | 83.7 | 56.4 | 50.9 | 62.6 | 57.0 |
| OpenScene [31] | ✓ | CLIP | 20.1 | 13.1 | 11.1 | 4.4 | 13.2 | 6.5 |
| LLM-Grounder [49] | ✓ | GPT-4 turbo | - | - | - | - | 17.1 | 5.3 |
| ZSVG3D [54] | ✓ | GPT-4 turbo | 63.8 | 58.4 | 27.7 | 24.6 | 36.4 | 32.7 |
| VLM-Grounder [48] | ✓ | GPT-4o | 66.0 | 29.8 | 48.3 | 33.5 | 51.6 | 32.8 |
| SeeGround [22] | ✓ | Qwen2-VL-72B | 75.7 | 68.9 | 34.0 | 30.0 | 44.1 | 39.4 |
| CSVG [52] | ✓ | Mistral-Large-2407 | 68.8 | 61.2 | 38.4 | 27.3 | 49.6 | 39.8 |
| **SPAZER (Ours)** | ✓ | GPT-4o | **80.9** | **72.3** | **51.7** | **43.4** | **57.2** | **48.8** |

**Evaluation metrics.** The ScanRefer dataset directly evaluates the accuracy of localizing the 3D bounding box of the object described in the query. The measuring metrics are annotated as Acc@0.25 and Acc@0.5, which represent the proportion of output bounding boxes whose IoU with the ground truth exceeds 0.25 or 0.5, respectively. In comparison, the Nr3D benchmark offers ground truth 3D bounding boxes for all objects, and the evaluation focuses on the selection accuracy.

**Implementation details.** Our agent adopts GPT-4o as the default VLM. The number of views $n$ is set to 4, and the Top-$k$ parameter is set to $k = 4$. For ScanRefer dataset, we follow prior works [54, 22] and use a pre-trained model [34] to obtain the 3D bounding boxes. All ablation studies are conducted on the same subset of Nr3D as [48]. Additional implementation details are included in the Appendix.

## 4.2 Quantitative comparison

**ScanRefer.** Tab. 1 shows the performance comparison on the ScanRefer dataset. Our SPAZER significantly outperforms all existing zero-shot methods, achieving the highest overall accuracy. Compared to previous state-of-the-art 2D-based method VLM-Grounder [48] and 3D-based method CSVG [52], SPAZER achieves gains of +24.4% / +16.0% and +7.6% / +8.9% in Acc@0.25 / Acc@0.5, respectively. In addition, our Acc@0.25 already matches or exceeds many fully-supervised 3DVG methods, such as ScanRefer [6], 3DVG-Transformer [57], and BUTD-DETR [16], without training on 3D datasets. This demonstrates strong zero-shot generalization. Although methods like Video-3D LLM [58] and GPT4Scene [33] show stronger performance, they require large-scale 3D-language data for extensive pre-training. In contrast, our method exhibits ease of deployment, and already achieves comparable Acc@0.25, showcasing its potential for practical application.

**Nr3D.** As shown in Tab. 2, our SPAZER surpasses all existing zero-shot methods in overall accuracy and even approaches the performance of fully-supervised methods, demonstrating the effectiveness of our spatial-semantic progressive reasoning framework. It is important to note that the default setting of the Nr3D dataset provides ground-truth 3D bounding boxes (without class label), meaning that the evaluation focuses primarily on the model's ability to reason over language to identify the correct object, rather than assessing the 3D localization precision. Under this setting, SPAZER achieves an overall accuracy of 63.8%, significantly outperforming prior zero-shot approaches such as VLM-Grounder [48] and EaSe [27]. Moreover, recent methods like CSVG [52] and Transcrib3D [10] use additional ground-truth object classes of 3D bounding boxes during inference. In this case, our method also achieves superior overall accuracy, especially with significant gains in the Hard category.

## 4.3 Qualitative comparison

Fig.3 illustrates qualitative grounding results of SPAZER in comparison with the representative 2D-based method VLM-Grounder[48] and 3D-based method SeeGround [22]. In (a), multiple identical objects (tables) appear in the scene, and **spatial cues** in the query (*e.g.*, "in the middle") are required to correctly localize the target. VLM-Grounder fails to resolve the spatial ambiguity due to its limited 3D understanding. In (b), the query includes **fine-grained semantics** (*e.g.*, "a white lamp on top of it"), which are hard to capture from 3D point cloud data alone, leading SeeGround to mispredict the target. In contrast, our SPAZER combines 3D spatial reasoning with 2D semantic verification,

Table 2: Quantitative comparison with supervised and zero-shot 3DVG methods on **Nr3D** [1].

| Method | Zero-shot | LLM/VLM | GT object class | Easy | Hard | Dep. | Indep. | Overall |
|---|---|---|---|---|---|---|---|---|
| ReferIt3DNet [1] | ✗ | - | ✗ | 43.6 | 27.9 | 32.5 | 37.1 | 35.6 |
| 3DVG-Transformer [57] | ✗ | - | ✗ | 48.5 | 34.8 | 34.8 | 43.7 | 40.8 |
| ViL3DRel [8] | ✗ | - | ✗ | 70.2 | 57.4 | 62.0 | 64.5 | 64.4 |
| 3D-VisTA [59] | ✗ | - | ✗ | 72.1 | 56.7 | 61.5 | 65.1 | 64.2 |
| MiKASA [5] | ✗ | - | ✗ | 69.7 | 59.4 | 65.4 | 64.0 | 64.4 |
| SceneVerse [17] | ✗ | - | ✗ | 72.5 | 57.8 | 56.9 | 67.9 | 64.9 |
| ZSVG3D [54] | ✓ | GPT-4 turbo | ✗ | 46.5 | 31.7 | 36.8 | 40.0 | 39.0 |
| SeeGround [22] | ✓ | Qwen2-VL-72B | ✗ | 54.5 | 38.3 | 42.3 | 48.2 | 46.1 |
| VLM-Grounder [48] | ✓ | GPT-4o | ✗ | 55.2 | 39.5 | 45.8 | 49.4 | 48.0 |
| EaSe [27] | ✓ | GPT-4o | ✗ | - | - | - | - | 52.9 |
| **SPAZER (Ours)** | ✓ | GPT-4o | ✗ | **68.0** | **58.8** | **59.9** | **66.2** | **63.8** |
| CSVG [52] | ✓ | Mistral-Large-2407 | ✓ | 67.1 | 51.3 | 53.0 | 62.5 | 59.2 |
| Transcrib3D [10] | ✓ | GPT-4 | ✓ | **79.7** | 60.3 | 60.1 | 75.4 | 70.2 |
| EaSe [27] | ✓ | GPT-4o | ✓ | - | - | - | - | 67.8 |
| **SPAZER (Ours)** | ✓ | GPT-4o | ✓ | 76.5 | **69.3** | **65.6** | **77.9** | **73.2** |

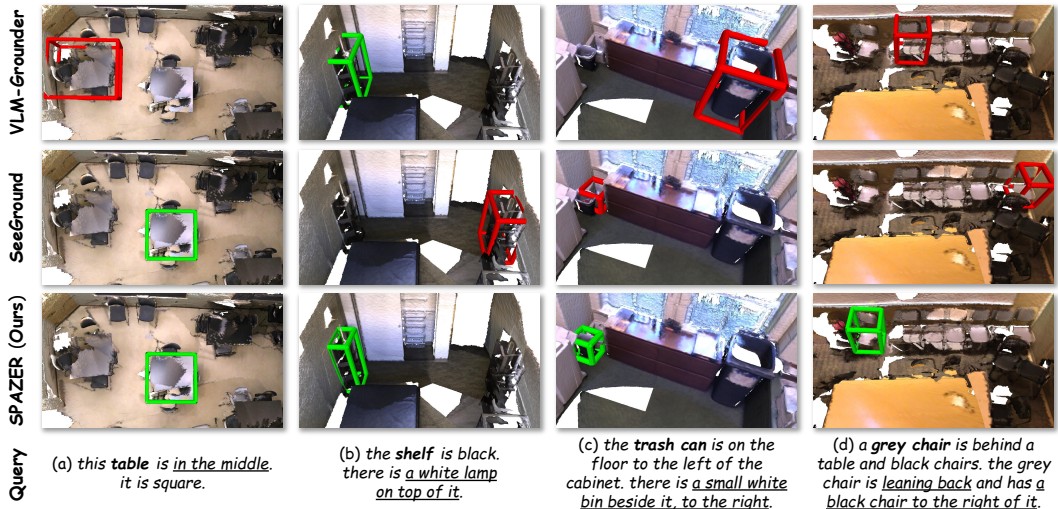

(a) this **table** is in the middle. it is square.

(b) the **shelf** is black. there is a white lamp on top of it.

(c) the **trash can** is on the floor to the left of the cabinet. there is a small white bin beside it, to the right.

(d) a **grey chair** is behind a table and black chairs. the grey chair is leaning back and has a black chair to the right of it.

Figure 3: Qualitative comparison of 3DVG results. Incorrectly predicted 3D bounding boxes are highlighted in red, while correct ones are in green. Key spatial and semantic cues are underlined.

allowing accurate object identification in both cases. Further, in more complex cases like (c) and (d), where both spatial and semantic cues must be jointly reasoned over (*e.g.*, "small white bin beside it", "leaning back", "black chair to the right of it"), SPAZER clearly outperforms both baselines by effectively leveraging the complementary strengths of both 3D geometry and 2D semantics.

### 4.4 Ablation study

**Effect of 3D holistic view selection (HVS).** As mentioned in Sec. 3.2, our SPAZER first holistically renders multiple perspective views and identifies the optimal one for the query text. To analyze the effectiveness of such design, we compare the following two variants: **(1) BEV**: Skip view selection and use the BEV by default, which is also consistent with prior work [33]. **(2) Random**: Use a random view instead of agent-selected one. In Tab. 3, BEV performs poorly as it may fail to reliably capture the target object. Random performs slightly better than BEV. In contrast, our agent leverages spatial understanding to select more informative views for subsequent steps, leading to improved overall accuracy (+3.6% vs. Random and +6.3% vs. BEV in average).

**Effect of candidate object screening.** At the candidate screening stage (Sec. 3.3), irrelevant object anchors are filtered out. As shown in Tab. 3, incorporating our proposed Retrieval-augmented anchor filtering (RAF) on top of the text-based baseline consistently improves accuracy by 2% to 7%. This is because text-based matching is vulnerable to inaccurate object class prediction, whereas our RAF introduces the visual object table as a complementary cue, leading to improved robustness.

Table 3: Ablation study on the component design of our agent. The default setting is highlighted.

| View Selection | | | Anchor Filtering | | w/o **JDM** (3D only) | | | | w/ **JDM** (3D+2D) | | | |
|---|---|---|---|---|---|---|---|---|---|---|---|---|
| **BEV** | **Random** | **HVS** | **Text-based** | **RAF** | # | **Hard** | **Dep.** | **Overall** | # | **Hard** | **Dep.** | **Overall** |
| ✓ | ✗ | ✗ | ✓ | ✗ | (a) | 36.8 | 37.5 | 41.2 | (g) | 48.3 | 49.0 | 53.2 |
| ✗ | ✓ | ✗ | ✓ | ✗ | (b) | 36.0 | 39.6 | 44.4 | (h) | 51.8 | 49.0 | 54.8 |
| ✗ | ✗ | ✓ | ✓ | ✗ | (c) | 38.6 | 42.7 | 46.8 | (i) | 52.6 | 53.1 | 56.4 |
| ✓ | ✗ | ✗ | ✓ | ✓ | (d) | 34.2 | 36.5 | 43.2 | (j) | 47.4 | 51.0 | 56.8 |
| ✗ | ✓ | ✗ | ✓ | ✓ | (e) | 31.6 | 41.7 | 46.8 | (k) | 50.9 | 55.2 | 59.2 |
| ✗ | ✗ | ✓ | ✓ | ✓ | (f) | **43.9** | **49.0** | **52.4** | (l) | **58.8** | **59.9** | **63.8** |

**Effect of 3D-2D joint decision-making (JDM).** The JDM module in Sec. 3.4 serves as the core of our approach, as it enables the integration of 3D spatial reasoning and 2D semantic verification within our agent. To enable a comprehensive analysis, in Tab. 3, we further include comparisons between w/o JDM and w/ JDM under each of the previously defined ablation settings. Here, w/o JDM refers to our agent making decisions using **3D-only** input, which is same as previous 3D-based methods [54, 22]. In this setting, our method achieves an overall accuracy of **52.4**%, outperforming ZSVG3D [54] and SeeGround [22] by $13.4\%$ and $6.3\%$, respectively. With the introduction of informative camera views as complementary 2D semantic context through camera view mapping, our agent achieves a substantial improvement—an average accuracy gain of **11.6%**. This result strongly supports the effectiveness of our spatial-semantic joint reasoning.

**Effect of $k$ in Top-$k$.** In the candidate screening stage, our agent gives the Top-$k$ most possible target object IDs. We evaluate the Top-$k$ accuracy (*i.e.*, whether the ground-truth object is in the $k$ candidates) and the overall grounding accuracy under different values of $k$. As shown in Fig. 4 (a) As $k$ increases, the Top-$k$ accuracy steadily improves, indicating a higher likelihood of including the ground-truth object into candidates. This also contributes to the improvement of overall accuracy. However, increasing $k$ from 4 to 5 leads to a performance drop, suggesting that an excessively large $k$ may introduce distracting candidates for subsequent decision-making.

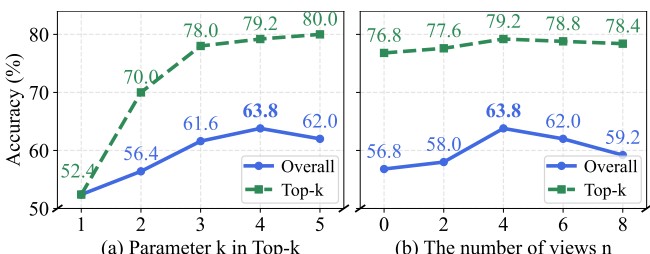

Figure 4: Ablation studies on parameter $k$ and $n$.

**Number of rendered views $n$.** In Fig. 4 (b), we show how varying the number of rendered views $n$ affects the accuracy. When $n = 0$, we only use the BEV by default. As $n$ increases from 0 to 4, the accuracy gradually improves. This is because more global views are provided for the agent to select from, allowing better observation of the target object. However, as $n$ further increases from 4 to 8, the accuracy starts to decline due to increasing overlap between views, which introduces redundant information and adds unnecessary noise, ultimately impairing the agent's decision-making. The Top-$k$ accuracy follows a similar trend to the overall accuracy, albeit with smoother variations. This suggests that even when the target object is included in the candidates, suboptimal 3D global view can still hinder the subsequent 3D-2D joint reasoning, highlighting the importance of view selection.

**Influence of VLMs.** The VLM serves as the "brain" of our agent, enabling effective query text interpretation and spatial-semantic reasoning. To investigate the impact of the adopted VLM on overall agent performance, we conduct a comparative analysis of several models, including proprietary ones (GPT-4o-mini, GPT-4o, GPT-4.1) and open-source alternatives (Qwen2-VL-72B [42], Qwen2.5-VL-72B [2]). Tab. 4 leads to the following conclusions: (1) The GPT series models exhibit stronger reasoning capabilities on the 3DVG task than open-source models; and (2) stronger VLMs contribute to improved performance of our agent, suggesting that continued advancements in

Table 4: Ablation study of the VLM used in our agent.

| VLM | Overall |
|---|---|
| Qwen2-VL-72B | 53.6 |
| Qwen2.5-VL-72B | 56.0 |
| GPT-4o-mini | 46.4 |
| GPT-4o | 63.8 |
| GPT-4.1 | **64.3** |

VLMs may further unlock its potential. Note that although GPT-4.1 can further boost the performance of our agent, we still adopt GPT-4o as the default VLM to ensure a fair comparison with previous methods.

Table 5: Performance comparison on the complete set and subset of the Nr3D [1] dataset.

| Dataset | Method | VLM | Easy | Hard | Dep. | Indep. | Overall |
|---------|--------|-----|------|------|------|--------|---------|
| Complete | SeeGround [22] | Qwen2-VL-72B | 54.5 | 38.3 | 42.3 | 48.2 | 46.1 |
| | Ours | Qwen2-VL-72B | 59.3 | 44.8 | 46.3 | 54.9 | 51.8 |
| | Ours | Qwen2.5-VL-72B | 62.4 | 46.9 | 49.9 | 56.8 | 54.3 |
| Subset | SeeGround [22] | Qwen2-VL-72B | 51.5 | 37.7 | 44.8 | 45.5 | 45.2 |
| | Ours | Qwen2-VL-72B | 62.5 | 43.0 | 51.0 | 55.2 | 53.6 |
| | Ours | Qwen2.5-VL-72B | 60.3 | 50.9 | 54.2 | 57.1 | 56.0 |

Table 6: Inference time of each step in our SPAZER. Ours significantly outperforms VLM-Grounder in efficiency. Both methods adopt GPT-4o as the VLM.

| Method | Step | Time (s) | Total (s) |
|--------|------|----------|-----------|
| Ours (SPAZER) | 3D Holistic View Selection | 5.2 | |
| | Candidate Object Screening | 8.5 | 23.5 |
| | 3D-2D Joint Decision-Making | 9.8 | |
| VLM-Grounder [48] | - | - | 50.3 |

**Performance on subset vs. complete dataset.** In consideration of budget and evaluation efficiency, we follow previous work VLM-Grounder [48] to evaluate our agent on the subset (250 selected samples) of each dataset. To further verify whether the results on the subset are comparable to those on the full dataset, we conduct additional experiments using open-source VLMs, as shown in Tab. 5. We observe that both our method and prior work SeeGround [22] exhibit consistent performance across the two dataset partitions, with overall accuracy variations under 2.0, which is negligible compared to the improvement achieved by our method.

**Inference time.** In Tab. 6, we break down the inference time of each step in SPAZER. The time consumption across different steps is relatively balanced, with the view selection step being faster since it requires no additional computation. Moreover, compared to VLM-Grounder [48], which also leverages 2D camera images for reasoning, our method achieves significantly higher inference efficiency. This is because we rely on VLM-selected anchors and require only a small number of images, avoiding the need to sample and filter all video frames.

## 5 Conclusion and limitations

In this work, we propose SPAZER, a novel spatial-semantic progressive reasoning agent for zero-shot 3D visual grounding. Instead of traditional object-level descriptions, SPAZER leverages holistic 3D rendered views to provide global spatial context, allowing VLMs to directly observe and interpret the 3D scene. It further employs a retrieval-augmented candidate screening strategy to enhance spatial reasoning and improve robustness against object category ambiguity. Extensive experiments verified the effectiveness of SPAZER, showcasing the potential of the 3D-2D joint reasoning paradigm in zero-shot 3DVG and its promise as a scalable alternative to supervised approaches.

**Limitations.** As we follow prior works and adopt pre-trained models to obtain 3D bounding boxes, the grounding accuracy will be related to their localization accuracy. In addition, the calculation of 3D-to-camera mapping can be affected by inaccurate camera parameters and depth information. For more detailed discussions of limitations, failure cases, and error types, please refer to the Appendix.

## Acknowledgments and Disclosure of Funding

This project is supported by the National Research Foundation, Singapore, under its NRF Professorship Award No. NRF-P2024-001.

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

# SPAZER: Spatial-Semantic Progressive Reasoning Agent for Zero-shot 3D Visual Grounding

## Appendix

**A. Implementation details**

**B. Additional experimental results and analysis:**

- B.1: Error type analysis
- B.2: Additional evaluation on cross-dataset generalization
- B.3: Stage-wise ablation on the effect of VLM capacity
- B.4: Case study on implied object grounding

**C. Limitations and border impact:**

- C.1: Analysis of failure cases
- C.2: Broader impact

## A Implementation details

**Computational resources.** Note that most of our experiments adopt GPT-4o as the VLM and it requires no GPU-based computation. The experiments involving Qwen2-VL-72B and Qwen2.5-VL-72B are conducted on multiple NVIDIA H100 GPUs.

**Model details.** The default VLM of our agent is GPT-4o (gpt-4o-2024-08-06). And the temperature is set to 0.2 to improve the reproducibility of the results. On ScanRefer dataset [6], we use Mask3D [34] to obtain 3D bounding box predictions, which is consistent with prior works [54, 22].

**Prompt design.** Our agent adopts several different prompts for the VLM. We first conduct target class prediction using the prompt in Tab. 7, which is similar with VLM-Grounder [48]. For **view selection (Sec. 3.2)**, we simply tell the VLM to select the view that can observe the query-described object most clearly using the prompt in Tab. 8. In **candidate object screening (Sec. 3.3)**, we prompt the VLM to select Top-$k$ candidate objects based on the annotated object IDs, as shown in Tab. 9. Eventually, **3D-2D joint decision-making (Sec. 3.4)** is achieved using the detailed prompt in Tab. 10.

Table 7: Prompt for reasoning the object category from the query description. `"{text}"` represents the input query.

---

**Prompt Template for Target Class Prediction**

You are working on a 3D visual grounding task, which involves receiving a query that specifies a particular object by describing its attributes and grounding conditions to uniquely identify the object.

Now, I need you to first parse this query and return the category of the object to be found. Sometimes the object's category is not explicitly specified, and you need to deduce it through reasoning. If you cannot deduce after reasoning, you can use `"unknown"` for the category. Your response should be formatted in JSON.

**Here are some examples:**

*Input:* Query: this is a brown cabinet. it is to the right of a picture.
*Output:*

```
{
    "target_class": "cabinet"
}
```

*Input:* Query: it is a wooden computer desk. the desk is in the sleeping area, across from the living room. the desk is in the corner of the room, between the nightstand and where the shelf and window are.
*Output:*

```
{
    "target_class": "desk"
}
```

...

**Now start your task:**
*Input:* `"{text}"`

---

Table 8: View selection prompt for identifying the best 3D view to locate the target object. `{target_class}` is the predicted object class and `"{text}"` denotes the query text.

> **Prompt Template for View Selection**
>
> You are good at finding the object in a 3D scene based on a given query description. These images show different views of a room. You need to find the `{target_class}` in this query description: `"{text}"`
>
> Please review all view images to find the target object and select the view that you can see the target object most clearly.
>
> **Output your answer in JSON format with these keys:**
>
> ```
> {
>   "reasoning": "Explain how you identified the target object,
>   and why you choose this view.",
>   "view": "2"  // The number of the view is in the top left
>   corner of the corresponding image.
> }
> ```

---

Table 9: Candidate screening prompt for identifying the Top-$k$ object IDs based on a given query. `{target_class}` is the predicted object class and `"{text}"` denotes the query text. `{n_topk}` is set to 4. `{object_id_list}` contains the valid object IDs after anchor filtering.

> **Prompt Template for Candidate Screening**
>
> Here is the annotated image of the selected view. All objects belonging to the `{target_class}` class are labeled by a unique number (ID) in red color on them.
> Please select the object ID that best matches the given query description: `"{text}"`
> Carefully analyze the specified conditions (such as shape, color, relative position with surrounding objects) in the given query, then select top-`{n_topk}` best-matched object IDs. The selected top-`{n_topk}` object IDs should be sorted in descending order of confidence. The object ID should be chosen from this list: `{object_id_list}`
>
> **Output your answer in JSON format with these keys:**
>
> ```
> {
>   "reasoning": "Explain how you identified and ranked the
>   top-{n_topk} target object IDs.",
>   "object_id": [1, 2, 3, 4, 5]  // A list of {n_topk}
>   selected target object IDs.
> }
> ```

Table 10: Input prompt for 3D-2D joint decision-making. {target_class} is the predicted object class and "{text}" denotes the query text. {object_id_list} contains the valid object IDs after anchor filtering.

---

**Prompt Template for 3D-2D Joint Decision-Making**

You are provided with a set of images depicting an indoor scene:

- A **global view image** showing the room's 3D layout from a fixed perspective.
- Several **camera images** captured from different viewpoints around the room.

All objects of interest in the scene are labeled with unique **object IDs** (in red), which are consistent across both the global and camera images.

Your task is to identify the object ID that best matches the given query description. Follow the steps below:

---

**1. Start with the global view image:**

- Analyze the overall spatial layout and object distribution in the room.
- Use the global view to evaluate **view-independent spatial relationships**, which do not rely on a specific viewpoint:
  *Examples include:* `near, close to, next to, far, above, below, under, on, top of, middle, opposite`

**2. Then examine the camera images:**

- Validate candidate objects identified from the global view.
- Evaluate **visual features**: color, shape, size, texture, and material.
- Use camera views to judge **view-dependent spatial relationships**, which depend on the camera perspective:
  *Examples include:* `left, right, in front of, behind, back, facing, looking, between, across from, leftmost, rightmost`
- **Tip:** Annotations may not always be at the center of the object. Focus on the full *spatial extent* and choose the ID that best represents the *main body* of the described object across both views.

**3. Iterate if needed:**

- If no candidate fully matches the query, return to the global view and reassess alternatives.
- Repeat verification with camera images until you confidently identify the best match.

---

**Task:**
Select the object ID of the target class: {target_class}
Query description: "{text}"
Object IDs to choose from: {object_id_list}

**Output format (JSON):**

```
{
  "reasoning": "Explain how you analyzed spatial relation-
  ships (view-dependent vs view-independent), cross-verified
  the object across views, and selected the best-matched
  ID.",
  "object_id": ID  // e.g., 10
}
```

# B    Additional experimental results and analysis

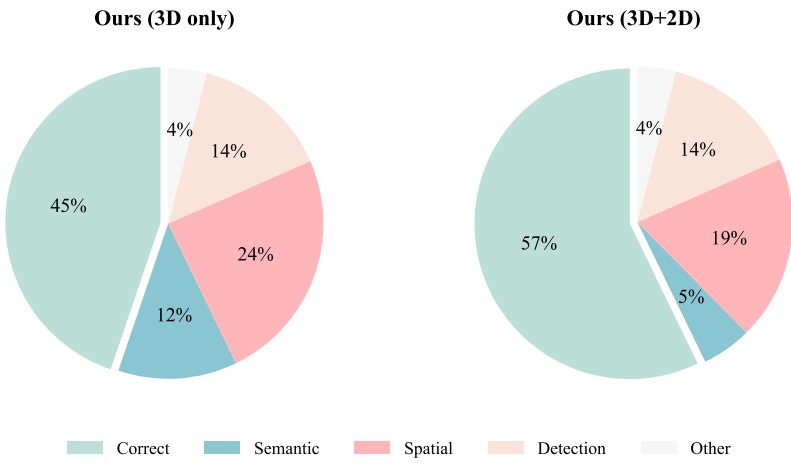

Figure 5: Error type distribution on ScanRefer dataset. Ours (3D only) indicates that our agent selects the target object directly at the candidate screening stage, without performing the subsequent 3D-2D joint decision-making. Ours (3D+2D) represents the full pipeline of our SPAZER method.

## B.1    Error type analysis

In Fig. 5, we present the distribution of different error types in SPAZER's predictions to provide insights into potential directions for further improvement. The main error types include: 1) Semantic: Errors caused by the model overlooking critical semantic cues, such as color, shape, *etc*.; 2) Spatial: Errors where the model fails to correctly interpret spatial relationships, including relative positions between objects or absolute directions (*e.g.*, northwestern-most); 3) Detection: Cases where the 3D detector fails to detect the target object or predicts the wrong category; 4) Other: Mainly due to referring ambiguities in the query text, where multiple objects in the scene could reasonably match the description based on human judgment.

**Uni-modal and multi-modal.** Compared to the 3D-only paradigm, incorporating 3D+2D significantly reduces errors in the Semantic category, indicating that 2D images provide important supplementary semantic cues for the agent. Additionally, the reliable view-dependent relational information from 2D images also helps reduce the occurrence of Spatial errors.

**Analysis and future work.** Based on the error type distribution of our SPAZER (right side of Fig. 5), Spatial errors account for the largest proportion, indicating that the primary challenge in 3DVG lies in understanding complex spatial relationships. To address this, we plan to further explore more effective 3D representations in future work. In addition, a considerable portion of errors is caused by the detector, suggesting that reducing the agent's reliance on the detector is one of the key issues to be addressed in future work.

Table 11: Cross-dataset evaluation on RIORefer. Both methods use the same detector trained on ScanRefer. SPAZER achieves superior zero-shot performance without any 3DVG-specific training.

| Method | Training Data | Setting | Acc@25 (%) | Acc@50 (%) |
|---|---|---|---|---|
| Cross3DVG [28] | ScanRefer | Supervised | 29.2 | 14.4 |
| **SPAZER (Ours)** | – | Zero-shot | **34.0** | **16.4** |

## B.2    Additional evaluation on cross-dataset generalization

To further verify the generalization ability of our SPAZER framework, we conducted additional experiments on the RIORefer benchmark [28], which is built upon 3RScan [40] and differs considerably from ScanNet [9] in terms of scene layout, object category distribution, and scanning conditions. This benchmark is therefore suitable for evaluating cross-dataset robustness and zero-shot transfer

Table 12: Stage-wise ablation on VLM capacity. We replace GPT-4o with GPT-4o-mini at one reasoning stage at a time to measure sensitivity.

| Experiment | Stage 1: View Selection | Stage 2: Anchor Filtering | Stage 3: Final Reasoning | Overall Acc (%) |
|---|---|---|---|---|
| A (SPAZER) | GPT-4o | GPT-4o | GPT-4o | 63.8 |
| B | GPT-4o-mini | GPT-4o | GPT-4o | 60.4 ($\downarrow$3.4) |
| C | GPT-4o | GPT-4o-mini | GPT-4o | 57.2 ($\downarrow$6.6) |
| D | GPT-4o | GPT-4o | GPT-4o-mini | 50.4 ($\downarrow$13.4) |

capability. In this experiment, both SPAZER and the supervised baseline (Cross3DVG) adopt the same 3D detector trained on ScanRefer to ensure a fair comparison. Notably, Cross3DVG is trained in a supervised manner with language–object annotations from ScanRefer, whereas SPAZER operates in a zero-shot setting without any 3DVG-specific training.

As shown in Tab. 11, SPAZER surpasses the supervised baseline even when evaluated across domains, demonstrating strong cross-dataset generalization and robustness to distribution shifts. This result highlights the adaptability of our framework, which effectively transfers knowledge from open-world vision–language models (VLMs) to unseen datasets without fine-tuning. We attribute SPAZER's superior cross-dataset performance to its reliance on the VLM's broad vision–language prior, instead of dataset-specific supervision. This design enables SPAZER to understand diverse object appearances and handle natural-language queries robustly, even under unfamiliar scene conditions.

## B.3 Stage-wise ablation on the effect of VLM capacity

To further analyze how SPAZER's performance depends on the reasoning capability of the Vision-Language Model (VLM) at different reasoning stages, we conducted **stage-wise ablation experiments**. Specifically, we replaced GPT-4o with a smaller model (GPT-4o-mini) at one stage at a time, while keeping the others unchanged. The results are summarized in Table 12. As shown in Table 12, the **final reasoning stage (Stage 3: 3D–2D joint decision-making)** is the most sensitive to the VLM's capacity, showing a notable $13.4\%$ drop when GPT-4o is replaced with GPT-4o-mini. This is expected, as this stage requires fine-grained spatial-semantic reasoning and cross-view understanding to accurately identify the target. The **anchor filtering stage (Stage 2)** experiences a moderate degradation ($\downarrow 6.6\%$), since its objective is to shortlist a set of likely candidates. Even with a weaker VLM, the correct anchor may still appear in the top-$k$ shortlist. In contrast, the **view selection stage (Stage 1)** is relatively robust ($\downarrow 3.4\%$). Because the same object can often be captured from multiple valid viewpoints, this stage places lower demands on fine-grained reasoning accuracy, making it less sensitive to VLM degradation. Overall, these results confirm that SPAZER follows a **coarse-to-fine progressive reasoning** paradigm. Early stages (view selection and anchor filtering) involve simpler reasoning steps and show higher tolerance to variations in VLM capability. In contrast, the final grounding stage requires precise spatial-semantic alignment and thus exhibits stronger dependency on the VLM's reasoning power.

## B.4 Case study on implied object grounding

Our proposed SPAZER is capable of handling implied object grounding, where the referred target is not explicitly mentioned but can be inferred from contextual cues. Unlike traditional text-matching–based approaches that rely solely on literal word overlap between query text and object class labels, our method integrates visual–semantic reasoning within the VLM and an anchor filtering mechanism guided by the VLM-inferred target class. This design allows the model to interpret implied intents (e.g., "something to drink") and associate them with relevant visual entities (e.g., bottles, cups, or glasses) even when such categories are not directly stated in the text.

As illustrated in Fig. 6, given the query "I am thirsty. Please bring me something from my desk," the model first identifies the most informative view of the desk and then filters objects related to drinking, such as bottles or cups. Among them, object 37 (a bottle) is consistently observed across multiple views and best matches the query semantics, while others (e.g., 66, 77, 1) are filtered out. This case study demonstrates that our reasoning-guided filtering effectively bridges explicit textual grounding and implicit semantic inference. By combining lightweight text-matching for efficient anchor pruning with VLM-based reasoning for category understanding, the model can infer the intended target even when the query describes an implied object. Consequently, it achieves robust performance on complex queries where conventional matching strategies fail to capture the latent semantics of user intent.

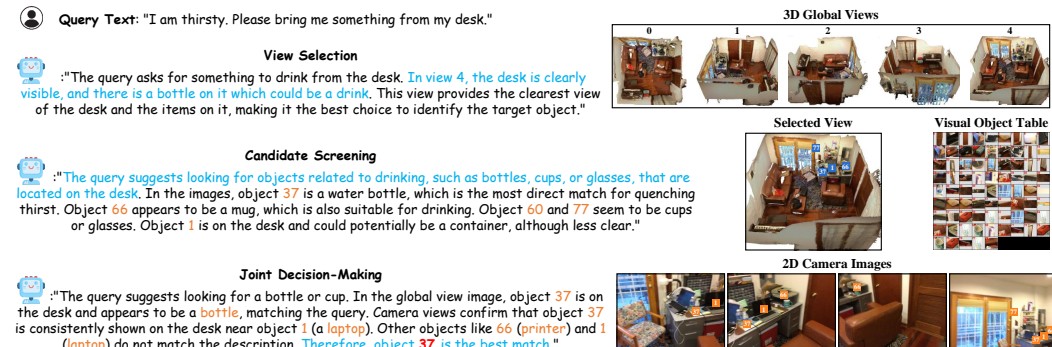

**Query Text**: "I am thirsty. Please bring me something from my desk."

**3D Global Views**

### View Selection

:"The query asks for something to drink from the desk. In view 4, the desk is clearly visible, and there is a bottle on it which could be a drink. This view provides the clearest view of the desk and the items on it, making it the best choice to identify the target object."

**Selected View**    **Visual Object Table**

### Candidate Screening

:"The query suggests looking for objects related to drinking, such as bottles, cups, or glasses, that are located on the desk. In the images, object 37 is a water bottle, which is the most direct match for quenching thirst. Object 66 appears to be a mug, which is also suitable for drinking. Object 60 and 77 seem to be cups or glasses. Object 1 is on the desk and could potentially be a container, although less clear."

**2D Camera Images**

### Joint Decision-Making

:"The query suggests looking for a bottle or cup. In the global view image, object 37 is on the desk and appears to be a bottle, matching the query. Camera views confirm that object 37 is consistently shown on the desk near object 1 (a laptop). Other objects like 66 (printer) and 1 (laptop) do not match the description. Therefore, object 37 is the best match."

Figure 6: Case study on implied object grounding by SPAZER.

# C   Limitations and border impact

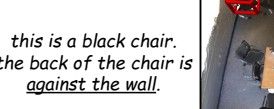
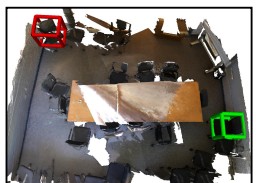
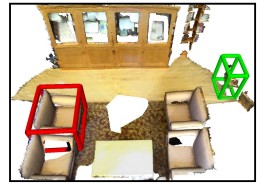

*this is a black chair. the back of the chair is against the wall.*

*the chair is south of the northwestern-most bookshelf. the chair is gray and has four legs.*

(a) Spatial relation

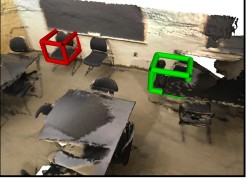
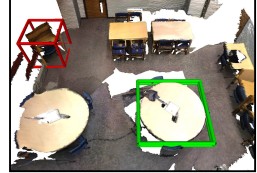

*the gray padded chair is on rollers. it is the only rolling chair at the table.*

*it is a white table. the white table is sitting behind the two brown desks in the far left corner of the room.*

(b) Semantic detail

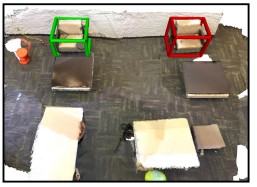
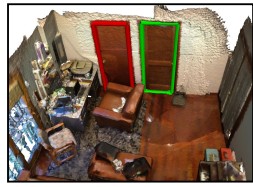

*the chair is in one of the sides of the room. the color of the chair is brown.*

*this is a brown door. it is across from an armchair.*

(c) Other (referring ambiguity)

Figure 7: Typical types of failure cases. The prediction and ground-truth are highlighted in red and green, respectively. (a) Relation error includes relative relation (e.g., against the wall) and absolute relation (e.g., northwestern-most). (b) Semantic error mainly involves detailed object attributes, such as shape (on rollers), color (white), material, etc. (c) Other errors are primarily caused by the referring ambiguity, i.e., multiple objects in the scene satisfy the query.

## C.1   Analysis of failure cases

Based on Fig. 5, there are mainly four types of failure cases. Since we adopt a detect-and-match paradigm similar to previous works [54, 22], detection-related errors are currently unavoidable. In future work, we plan to explore how to enable the agent to directly produce localization results. The remaining three types of errors are illustrated through case studies in Fig. 7.

Regarding spatial relations, failures mainly occur in: 1) complex positional relationships, which often involve both the orientation of the target object and its relation to surrounding objects; 2) directional

terms (e.g., south, northwest). In future research, we plan to incorporate visual prompts indicating orientation into the 3D representation.

For semantic details, when the scene contains multiple visually similar objects, the candidate screening stage may fail to include the correct target into the Top-$k$ list, preventing effective semantic verification in subsequent steps. This reflects a limitation of our current method.

Lastly, we observed that some samples in the dataset exhibit referring ambiguity. As shown in Fig. 7(c), both the predicted result and the ground truth can satisfy the query description based on human interpretation. Therefore, the construction of higher-quality 3DVG datasets stands out as a critical challenge that needs to be addressed in future research.

## C.2   Broader impact

Our VLM-driven agent SPAZER for 3D visual grounding offers potential benefits in areas such as human-robot interaction, augmented reality, and assistive technologies by enabling more intuitive object localization from language. However, it may also carry risks, such as biases inherited from pre-trained models, which could affect performance in diverse environments. Future work should address these concerns through fairness-aware training, improved interpretability, and responsible deployment.

