# OpenReview forum: "SPAZER: Spatial-Semantic Progressive Reasoning Agent for Zero-shot 3D Visual Grounding"
_NeurIPS.cc/2025/Conference — NeurIPS 2025 poster_

### Official Review · Reviewer_e7pa · 2025-06-27

**Clarity:** 3
**Significance:** 3
**Originality:** 3
**Rating:** 5
**Confidence:** 4

**Summary:**

This paper proposes a new zero-shot 3D visual grounding technique, titled SPAZER, which leverages VLMs to output in text the 3D bounding box coordinates of the object which is referred in the language prompt. To achieve this, the method first identifies the set of candidate objects from an optimally selected 3D view of the scene. From this 3D view and from a set of selected 2D images that clearly depict the candidates, the method then selects and localizes the referred object. This allows joint reasoning about the 3D configuration of objects in the scene and fine-grained semantic cues, which are both crucial for the task at hand. The authors validate SPAZER on the two major 3D visual grounding benchmarks, i.e. ScanRefer and Nr3D, showing top performance among zero-shot methods that do not train explicitly on 3D data.

**Questions:**

1. Can the authors provide some efficiency comparison of the method to competing approaches? Relevant quantities are latency, parameter efficiency, and FLOPS.

2. Can the authors provide a more in-depth ablation and examination of the effect of the particular VLM which is employed by their method on the visual grounding performance? Is the method more vulnerable / more robust to VLM failures in the 3D viewpoint selection stage, the candidate selection stage, or the final grounding stage?

**Ethical Concerns:**

["NO or VERY MINOR ethics concerns only"]

**Final Justification:**

The paper examines an important problem in a highly relevant, zero-shot setting. The proposed pipeline is novel and constitutes a creative utilization of state-of-the-art VLMs to marry them with traditional visual recognition models for 3D grounding. Finally, the experimental comparisons are comprehensive and thorough and they demonstrate a state-of-the-art performance in the task at hand. The rebuttal by the authors has addressed my concerns with respect to efficiency comparisons, effect of selected VLM on the performance of the pipeline, and missing related works, which is why I have increased my rating to 5: Accept.

**Limitations:**

Yes.

**Paper Formatting Concerns:**

None.

**Quality:**

2

**Strengths And Weaknesses:**

Strengths:

1. The examined problem and setting is important, as natural language needs to be integrated with the 3D environment for better understanding in embodied applications. Also, the zero-shot setting is relevant and interesting.

2. The proposed pipeline of SPAZER is novel and constitutes a creative utilization of state-of-the-art VLMs to marry them with traditional visual recognition models.

3. The experimental comparisons are comprehensive and thorough and they demonstrate a state-of-the-art performance in 3D visual grounding. The method is also ablated in detail.

Weaknesses:

1. No comparison or analysis is conducted on the computational efficiency of the method. It is not clear whether the multi-step pipeline of SPAZER, which i.a. includes at least 3 successive evaluations of a bulky VLM such as GPT 4o, still has comparable latency to competing VLM-based approaches or to traditional supervised methods, some of which are more accurate than SPAZER.

2. While the examined task resides in 3D, the proposed method only operates on 2D views. This has been the choice of previous works too, but I still wonder whether the explicit reasoning about 3D from these 3D views, e.g. via estimating their depth maps, could easily lead to better localization of 3D bounding boxes, as this technique has proven effective in the related context of 3D object detection from cameras.

3. The obtained results largely depend on the performance of the vision-language model which is used. For example, in Fig. 4, performance degrades for more than 4 views, even though for more views the information presented to the model is more and can in principle lead to better results. Is this performance deterioration for more generated views a limitation of the employed VLM or a fundamental limitation of the proposed method?

4. While the related work is surveyed quite well, there are still important works which are missing, especially from the traditional, supervised 3D visual grounding line, e.g. [A], [B], [C] - the latter work in particular also employs multi-view 2D images from 3D for visual grounding. The authors should include a few additional works in the literature.

[A] Text-guided graph neural networks for referring 3D instance segmentation. In AAAI, 2021.

[B] Four Ways to Improve Verbo-visual Fusion for Dense 3D Visual Grounding. In ECCV, 2024.

[C] Multi3Drefer: Grounding text description to multiple 3d objects. In ICCV, 2023.

---

> ### Author Rebuttal · Authors · 2025-07-30
>
> ## W1 & Q1: The computational efficiency of SPAZER.
> We thank the reviewer for the suggestion. In the Table below, we report the inference time (latency) of each component in SPAZER and compare it against VLM-Grounder, which also uses GPT-4o as the VLM. Our method demonstrates significantly better inference efficiency (23.5s vs. 50.3s). Since we identify candidate objects using 3D global views, we only require **a small number (k) of informative camera images** for further reasoning, thus avoiding the need to exhaustively sample and filter all video frames.
>
> | Method                   | Step                                             | Time (s) | Total (s) |
> |---------------------------|--------------------------------------------|-------------|-------------|
> | **Ours (SPAZER)** | 3D Holistic View Selection          | 5.2         |               |
> |                                 | Candidate Object Screening       | 8.5         |               |
> |                                 | 3D–2D Joint Decision-Making     | 9.8         | **23.5**  |
> | VLM-Grounder        | -                                                   | -             | **50.3**  |
>
> Note that the cost of 3D rendering and 3D-to-camera projection is almost negligible compared to the VLM inference, which dominates the total runtime. Since SPAZER is entirely training-free, it **does not introduce any additional trainable parameters** beyond the underlying VLM. Therefore, parameter efficiency is primarily determined by the VLM used (e.g., GPT-4o), which is shared across compared methods.
>
> ## W2: Whether explicit 3D reasoning like depth estimation from 2D views could lead to better localization.
> We agree that incorporating explicit 3D reasoning—such as depth estimation or geometry-aware modules—has the potential to further improve localization accuracy, especially in complex or cluttered environments.
>
> But our current method is designed with a **training-free, zero-shot setting** in mind, following prior works like SeeGround and ZSVG3D. To ensure fair comparison, we avoid introducing additional modules that require supervised fine-tuning.
>
> In fact, **our framework inherently possesses 3D awareness**. It draws inspiration from how humans typically perform 3D visual grounding during annotation—by interactively navigating 3D scenes through rotation and multi-view observation (e.g., in tools like MeshLab or Open3D) until they can confidently localize the object described by language. This motivated us to design a multi-view rendering and reasoning pipeline that enables the VLM to aggregate spatial and semantic cues from multiple perspectives. Although we do not explicitly estimate depth, this design equips the model with a degree of implicit 3D awareness, allowing it to reason about visibility, relative positioning, and occlusion.
>
> In future work, we are very interested in integrating explicit geometry cues, such as estimated depth maps or surface normals, to further enhance the 3D reasoning capability.
>
> ## W3: Why performance deterioration for more generated views.
> As shown in Fig. 4(b) of the manuscript, SPAZER achieves the best performance when using n = 4 views, while a slight performance drop is observed as the number of views increases beyond that. We believe this phenomenon is arises from the **viewpoint quality** instead of the number of views.
>
> When n = 4, the selected views are rendered from **top-oblique** angles around the room, offering broad and minimally occluded observations of the scene. However, as n increases, additional views are often captured from lower or corner angles, which tend to suffer from occlusions caused by walls, furniture, or self-occluding objects.
>
> To further investigate this, we conducted a **controlled experiment**: even with n=4, when all views were rendered specifically from the four room corners—i.e., the additional views introduced when increasing from n=4 to n=8—performance dropped to **~58.4% Acc@50**, compared to 63.8% using the top-oblique setting. These results highlight that spatial understanding is more influenced by view quality than by the number of views. Setting a large value of n may introduce too many low-quality or occluded views, which can overwhelm the model and lead to performance degradation.
>
> | View Setting           | Acc@50 (%) |
> |------------------------|-------------|
> | 2 views                | 58.0        |
> | 4 views (Top-oblique)  | **63.8**    |
> | 4 views (Room-corner)  | 58.4        |
> | 8 views                | 59.2        |
>
> Since this stage only requires the VLM to perform **coarse reasoning based on global views**, and high-quality close-up camera images will be incorporated in the subsequent stages, we believe introducing an excessive number of views is also unnecessary at this point.
>
> We will include this additional analysis and ablation in the revised version.
>
> ## Q2: More in-depth ablation of the effect of VLMs and sensitivity at different reasoning stages.
> To analyze how SPAZER’s performance depends on the reasoning capacity of the VLM at different stages, we conducted **stage-wise ablation experiments**, replacing GPT-4o with a smaller model (GPT-4o-mini) at one stage at a time. The results are summarized below:
>
> | Experiment | Stage 1: View Selection | Stage 2: Anchor Filtering | Stage 3: Final Reasoning | Overall Acc (%) |
> |------------|-------------|--------------|--------------|--------------|
> | A (SPAZER) | GPT-4o      | GPT-4o       | GPT-4o       | 63.8         |
> | B          | GPT-4o-mini | GPT-4o       | GPT-4o       | 60.4 (↓ 3.4) |
> | C          | GPT-4o      | GPT-4o-mini  | GPT-4o       | 57.2 (↓ 6.6) |
> | D          | GPT-4o      | GPT-4o       | GPT-4o-mini  | 50.4 (↓ 13.4)|
>
> As the results show:
> - The final stage (Stage 3: 3D–2D joint decision-making) is the most sensitive to VLM capability, with a **13.4%** drop when downgraded. This is expected, as this stage requires the VLM to reason across multiple views and resolve fine-grained spatial-semantic ambiguities to identify the target.
> - The anchor filtering stage (Stage 2) shows moderate degradation (**↓ 6.6%**), since its goal is to shortlist a set of likely candidates. Even under partial VLM failure, the target may still appear in the top-k shortlist.
> - The view selection stage (Stage 1) is relatively robust (**↓ 3.4%**). Since the same object can typically be observed from multiple valid perspectives, this stage imposes relatively lower demands on the VLM's fine-grained reasoning ability, making it less sensitive to moderate variations in VLM performance.
>
> These results confirm that our framework follows a **coarse-to-fine progressive reasoning** paradigm. **Early stages** like view selection and anchor filtering involve relatively simpler tasks and thus exhibit higher tolerance to VLM degradation. In contrast, the **final grounding stage** requires fine-grained spatial-semantic reasoning, making it more sensitive to the VLM's capacity.
>
> We thank the reviewer for the valuable suggestion, and we will include this analysis and ablation in the next version.
>
> ## W4: Inadequate related work.
> We thank the reviewer for highlighting this point and for recommending these valuable references.
> While we aimed to provide a comprehensive overview of the related literature, we acknowledge the omission of several important works from the supervised 3D visual grounding line. In particular, the suggested works:
>
> [A] Text-guided Graph Neural Networks for Referring 3D Instance Segmentation (AAAI 2021),
>
> [B] Four Ways to Improve Verbo-visual Fusion for Dense 3D Visual Grounding (ECCV 2024), and
>
> [C] Multi3DRefer: Grounding Text Description to Multiple 3D Objects (ICCV 2023)
>
> are highly relevant to our task and provide complementary perspectives, including graph-based reasoning and multi-view fusion.
> We will incorporate these references and revise the related work section to better position our method in the context of prior supervised approaches.

---

> > ### Comment · Reviewer_e7pa · 2025-08-05
> > **Comment on rebuttal by Reviewer e7pa**
> >
> > Dear authors,
> >
> > Thank you for your detailed and insightful response.
> >
> > **1. Efficiency comparisons**
> >
> > While the provided comparison to another VLM-based grounding model partially addresses my concern, I am still missing the comparison in terms of FLOPS to such a method, as well as latency + parameter count + FLOPS comparison to traditional supervised methods, which I expect to be able to run inference much faster than the proposed method.
> >
> > **2. Explicit reasoning about 3D**
> >
> > I understand the argument that the proposed model implicitly reasons about 3D, which means that the authors take a different path towards 3D grounding than explicit 3D-based approaches. This is a valid choice, even though it may appear unintuitive, given the strong generalization of big VLMs. It will be indeed a good idea for the authors to state explicit 3D reasoning with depth maps or surface normals as a potential direction for future work on improving their current approach.
> >
> > **3. Performance degradation with larger number of views**
> >
> > The provided comparison of different numbers of views as well as same numbers of views but from different viewpoints is enlightening. The limitation with more views seems to originate from the VLM and its difficulty in isolating the most informative views from larger sets. At the same time, the particular number of four views at which the performance culminates is intuitive when considering the rectangular layout of the indoor data used in the experiments. It will indeed be useful to include this ablation in the revised version.
> >
> > **4. Effect of VLM at different stages of the pipeline**
> >
> > The authors have provided a well-designed ablation to respond to this concern of mine. The results of this ablation indeed show the reduced sensitivity of the approach to limited VLM performance in the earlier stages of the pipeline and indicate that increasing accuracy of VLMs in the future will likely positively impact the visual grounding performance of pipelines similar to the proposed one. It will indeed be valuable to include this analysis in the revised version.
> >
> > **5. Related work**
> >
> > It is nice that the authors have acknowledged in their rebuttal the relevance of traditional, supervised 3D grounding works such as the indicative references. It will indeed be good to slightly revise the related work section to better compare and contrast the proposed method against such works.
> >
> > Overall, the rebuttal by the authors largely addresses my concerns and I am leaning towards improving my rating. However, I am still open at this point to hear the views of Reviewer and any remaining concerns they may have on the submission.

---

> > > ### Author Response · Authors · 2025-08-06
> > >
> > > We thank the reviewer for the constructive feedback and for acknowledging the value of our responses regarding 3D reasoning, multi-view ablation, VLM stage sensitivity, and related work. We will incorporate these analyses and clarifications into the revised version.
> > >
> > > ### Response to Point 1: Efficiency Comparisons
> > > The comparisons across latency, parameter count, and FLOPs, covering both SOTA supervised method (e.g., GPT4Scene) and VLM-based zero-shot method (e.g., VLM-Grounder) are summarized in the two tables below:
> > >
> > > ***Table 1.** Latency comparison.*
> > >
> > > | Method         | Zero-shot | VLM | Latency (s) |
> > > |----------------|-----------|-------------|-------------|
> > > | GPT4Scene     | ✗          | Qwen2-VL-7B         | 1–2         |
> > > | VLM-Grounder| ✓            | GPT-4o         | 50.3        |
> > > | SPAZER            | ✓          | GPT-4o         | 23.5        |
> > >
> > > SPAZER achieves significantly lower latency (23.5s) and FLOPs (~1.3×10¹⁵) than VLM-Grounder (50.3s, ~3.2×10¹⁵). As the parameters and FLOPs of GPT-4o are inaccessible, we provide further comparisons using open-sourced VLMs:
> > >
> > > ***Table 2.** Parameters and FLOPs comparison.*
> > >
> > > | Method         | Zero-shot | VLM | Parameters | FLOPs            |
> > > |----------------|-----------|------------|------------|------------------|
> > > | GPT4Scene      | ✗         |Qwen2-VL-7B| 7B         | ~1.0 × 10¹³      |
> > > | VLM-Grounder  | ✓         |Qwen2-VL-72B| 72B        | ~3.2 × 10¹⁵      |
> > > | SPAZER        | ✓         |Qwen2-VL-72B| 72B        | ~1.3 × 10¹⁵      |
> > >
> > >
> > > Supervised methods like GPT4Scene achieve faster inference (1–2s latency) and lower FLOPs (~1.0×10¹³) due to their task-specific design and training. In contrast, zero-shot methods rely on large VLMs to compensate for the lack of task-specific supervision, which leads to higher inference cost. However, zero-shot methods remove the need for costly data annotation and task-specific training.

---

> > > > ### Comment · Reviewer_e7pa · 2025-08-06
> > > >
> > > > Dear authors,
> > > >
> > > > Thank you for the additional comment. It addresses effectively my remaining concerns on the efficiency of the proposed method and puts this element into perspective well. The associated conclusions drawn in your latest response will be valuable for inclusion in a revised version.

---

### Official Review · Reviewer_2LyA · 2025-06-30

**Clarity:** 3
**Significance:** 3
**Originality:** 3
**Rating:** 5
**Confidence:** 5

**Summary:**

This paper presents SPAZER, a novel agent for zero-shot 3D visual grounding that integrates both spatial and semantic reasoning in a progressive framework.  SPAZER leverages a Vision-Language Model (VLM) to jointly reason over both modalities without 3D-specific training. The agent first performs a holistic analysis of the 3D scene by rendering multiple global views. Subsequently, it refines object identification through joint decision-making. SPAZER significantly outperforms existing zero-shot methods.

**Questions:**

1.  How sensitive is SPAZER’s performance to the quality and domain of these detectors? Can the authors provide quantitative analysis or qualitative discussion on the impact of noisy or misaligned detections?
2. The SPAZER framework involves multi-view 3D rendering, candidate screening, and multi-modal (3D-2D) reasoning, which may introduce computational and memory overhead. Can the authors clarify the computational cost?
3. Does the evaluation conduct on the whole val set or a subset?

**Ethical Concerns:**

["NO or VERY MINOR ethics concerns only"]

**Final Justification:**

The author has addressed my concern. Please include this content in the main manuscript, especially the full set evaluation.

**Limitations:**

yes

**Quality:**

3

**Strengths And Weaknesses:**

# Strengths
1. The paper proposes a well-motivated, coherent framework that effectively integrates both 3D spatial reasoning and 2D semantic verification through a progressive multi-stage pipeline.
2. The authors provide extensive quantitative and qualitative experiments on two widely-used benchmarks (ScanRefer and Nr3D), consistently demonstrating superior performance over both zero-shot and supervised baselines.
# Weaknesses
1. The approach still relies on the accuracy of off-the-shelf 3D object detection models to extract bounding boxes and class labels. Errors in these components may propagate through the pipeline.
2. The performance largely depend on the ablility of VLMs (Qwen 56 Acc vs 4o 63.8 Acc)

---

> ### Author Rebuttal · Authors · 2025-07-30
>
> ## W1 & Q1: Analysis on the impact of detection errors.
> We thank the reviewer for highlighting the importance of robustness to detection results. We designed a series of experiments to systematically analyze the impact of two common types of detection errors: (1) Bounding box category noise, and (2) Bounding box misalignment (position and size perturbation).
>
> **(1) Robustness to category noise**
>
> Thanks to the design of our Retrieval-augmented anchor filtering (RAF) module, SPAZER is able to reason over the visual appearance of each detected object instead of relying solely on the predicted class name. As shown in our experiments (see Table below), our method demonstrates **stronger robustness over category noise**—particularly under the extreme case where all predicted class labels are removed, SPAZER still maintains a reasonable level of accuracy. In contrast, previous methods that depend text-based 3D representations struggle or completely fail in such noisy settings due to their heavy dependence on accurate class labels.
>
> | Corruption Ratio     | SPAZER         | SeeGround         |
> |----------------------|----------------|-------------------|
> |                      | Acc@50         | Acc@50            |
> | 0% (clean)           | 48.8           | 37.6              |
> | 10%                  | 45.6           | 32.8              |
> | 20%                  | 39.6           | 28.4              |
> | 50%                  | 34.4           | 17.2              |
> | 100% (no cls label)  | 32.8           | 6.4               |
>
> **(2) Sensitivity to geometric perturbation**
>
> To assess the robustness of SPAZER to geometric misalignment, we simulate misaligned detection boxes by injecting Gaussian noise into both the position and size of each bounding box. The table below shows a clear performance drop as the perturbation strength increases.
> We acknowledge that SPAZER currently adopts detector-predicted 3D bounding boxes directly as anchors—this design choice stems from the fact that **current VLMs lack the ability to directly predict 3D bounding boxes**. As a result, SPAZER is naturally more sensitive to geometric perturbations. This limitation is shared with prior works (e.g., CSVG3D, SeeGround) that follow a similar paradigm, and we view improving robustness to box localization noise as a promising direction for future development.
>
> | (position, size) | Acc@25  | Acc@50  |
> |------------------|---------|---------|
> | (0.00, 0.00)     | 57.2    | 48.8    |
> | (0.05, 0.02)     | 48.0    | 35.6    |
> | (0.10, 0.05)     | 40.8    | 19.2    |
>
> We believe this is an important limitation and a valuable direction for future research, as stated in our paper (Sec. 5).  However, our method takes an **important step forward by explicitly addressing the robustness to noisy object class predictions**—an issue that previous works have largely overlooked.
> In the future work, we aim to explore more autonomous grounding approaches, possibly incorporating self-generated or self-corrected 3D anchors to reduce dependence on external detectors.
>
> ## Q2: The computational cost of SPAZER.
> In Table below, we break down the inference time of each step in SPAZER.
>
> | Method            | Step                             | Time (s) | Total (s) |
> |-------------------|----------------------------------|----------|-----------|
> | **Ours (SPAZER)** | 3D Holistic View Selection       | 5.2      |           |
> |                   | Candidate Object Screening       | 8.5      | **23.5**  |
> |                   | 3D–2D Joint Decision-Making      | 9.8      |  |
> | VLM-Grounder | -                                | -        | **50.3**  |
>
> The time consumption across different steps is relatively balanced, with the view selection step being faster since it requires no additional computation. Moreover, compared to VLM-Grounder, which also leverages 2D camera images for reasoning, our method achieves significantly higher inference efficiency. This is because we rely on VLM-selected anchors and require only a small number of images, avoiding the need to sample and filter all video frames.
>
> ## Q3: Evaluation on the whole val set or a subset?
> In consideration of budget and evaluation efficiency, we follow previous work VLM-Grounder to
> evaluate our agent on the **subset** (250 selected samples) of each dataset. To further verify whether the results on the subset are comparable to those on the whole val dataset, we conduct additional experiments using open-source VLMs, as shown in the table below. We observe that both our method and prior work SeeGround exhibit consistent performance across the two dataset partitions, with overall accuracy variations **< 2.0**, which is negligible compared to the improvement achieved by our method. This confirms the validity of using the subset for fair and meaningful comparison.
>
> | Dataset | Method             | VLM              | Easy | Hard | Dep. | Indep. | Overall |
> |---------|--------------------|------------------|------|------|------|--------|---------|
> | Whole val set    | SeeGround     | Qwen2-VL-72B     | 54.5 | 38.3 | 42.3 | 48.2   | 46.1    |
> |         | Ours               | Qwen2-VL-72B     | 59.3 | 44.8 | 46.3 | 54.9   | 51.8    |
> |         | Ours               | Qwen2.5-VL-72B   | 62.4 | 46.9 | 49.9 | 56.8   | 54.3    |
> | Subset  | SeeGround     | Qwen2-VL-72B     | 51.5 | 37.7 | 44.8 | 45.5   | 45.2    |
> |         | Ours               | Qwen2-VL-72B     | 62.5 | 43.0 | 51.0 | 55.2   | 53.6    |
> |         | Ours               | Qwen2.5-VL-72B   | 60.3 | 50.9 | 54.2 | 57.1   | 56.0    |
>
> ## W2: The performance largely depends on VLMs.
> Since the VLM serves as the "brain" of SPAZER—responsible for understanding the query text, reasoning about spatial relations, and inferring object attributes—we believe it is reasonable that the overall performance depends on the capability of the underlying VLM.
>
> However, even when using a relatively less powerful VLM like Qwen2-VL-72B, our method still outperforms prior approaches such as SeeGround that use the same VLM. This suggests that our spatial-semantic progressive reasoning pipeline contributes significantly beyond the choice of VLM itself.
>
> Furthermore, our framework is designed to be modular and model-agnostic, allowing easy substitution of the VLM component. As newer and stronger VLMs continue to emerge, our method can naturally benefit from these advances without requiring any task-specific training or architecture changes, making it **convenient to adapt in future deployments**.

---

> > ### Comment · Reviewer_2LyA · 2025-08-05
> >
> > The author has addressed my concern. Please include this content in the main manuscript, especially the full set evaluation.

---

> > > ### Author Response · Authors · 2025-08-05
> > >
> > > Thank you for your constructive suggestion! We will revise the manuscript accordingly and include the full set evaluation and relevant explanations in the main paper.

---

### Official Review · Reviewer_8wRM · 2025-07-02

**Clarity:** 3
**Significance:** 2
**Originality:** 2
**Rating:** 4
**Confidence:** 4

**Summary:**

The paper introduces SPAZER, a zero-shot 3D visual grounding agent designed to integrate 3D and 2D reasoning through a progressive reasoning framework. SPAZER first performs holistic 3D spatial analysis using multi-view BEV rendering, then employs a pre-trained 3D detector for coarse object localization, and finally uses 2D images for fine-grained semantic verification and joint decision-making. Experimental results on ScanRefer and Nr3D benchmarks show significant improvement over existing zero-shot methods.

**Questions:**

1. Can the authors explain how their framework addresses open-vocabulary tasks?

2. Can the authors explain their decision to rely on text-matching anchor filtering instead of LLM-based approaches? Could an LLM-driven approach mitigate the current limitations?

3. Can the authors provide examples and analyses of how their method outperforms existing methods in out-of-scannet scenarios?

**Ethical Concerns:**

["NO or VERY MINOR ethics concerns only"]

**Final Justification:**

The experiments on out-of-scannet experiments strengthen the paper's claim and other supplement experiments addressed most of my concerns.

**Limitations:**

yes

**Quality:**

2

**Strengths And Weaknesses:**

Strengths

1. The authors introduces a novel framework which intergrates 3D and 2D reasoning, addressing a the gap in existing visual grounding agents.

2. The performance on ScanRefer and Nr3D significantly outperforms prior zero-shot methods.

3. The authors conduct detailed ablation studies, clearly demonstrating the contributions of each component.

Weaknesses

1. Although multiple views are rendered, they are essentially variants of BEV. This approach inherently fails in certain scenarios, like scenes with ceilings and multi-floor buildings, significantly limiting its applicability.

2. This method heavily relies on a pre-trained 3D detector, fundamentally restricting the upper-bound capabilities. The 3D detector before VLM systematically harms its performance in open-vocabulary tasks. Furthermore, existing detectors like Mask3D have not claimed zero-shot abilities, conflicting with the zero-shot setting in this work.

3. The anchor filtering mechanism is based on text matching algorithms rather than LLMs. Consequently, it cannot effectively identify implied object targets (e.g., "I'm thirsty, please bring me something from my desk."). Although a switching mechanism is proposed, it merely acts as a remedial measure post-failure, not resolving the fundamental issue.

4. The experiments are conducted on ScanNet-based benchmarks, which mitigates the above issues. I strongly recommend the authors to conduct broader evaluations in diverse scenarios.

5. The authors appear not to have uploaded their supplementary material correctly.

---

> ### Author Rebuttal · Authors · 2025-07-30
>
> ## W1: Applicability in scenes with ceilings and multi-floor buildings.
> We thank the reviewer for highlighting this important point.
>
> For scenarios with **ceilings**, our current method already addresses this by removing points above a height threshold relative to the scene’s maximum height. Specifically, we filter out the top 0.4 meters to eliminate ceiling points before rendering.
> We find this strategy to be simple yet effective in our experiments (Acc@25: **57.2** w/ ceiling removal vs. **52.4** w/o).
>
> For **multi-floor** scenarios, we acknowledge that our current implementation does not explicitly distinguish floor levels. However, the commonly used benchmarks (e.g., ScanRefer, Nr3D, RIORefer) predominantly feature single-floor indoor scenes, making this less of a concern in our current setting.
> And we believe our framework **can be extended** to better support multi-floor scenarios. For instance: a coarse floor segmentation based on height clustering or semantic segmentation could isolate floors; and the VLM could further leverage semantic cues from the query language (e.g., “second-floor desk”) to localize the relevant region. We appreciate the reviewer’s suggestion and consider this an important direction for future work.
>
> ## W2: The adopted detector conflicts with the zero-shot setting.
> It is correct that existing detectors like Mask3D are not designed for zero-shot detection. However, our setting focuses on zero-shot 3D visual grounding, which is a **fundamentally different task**: the goal is not to recognize object categories, but to identify the correct object in the 3D scene based on a free-form natural language query.
>
> Importantly, training a 3DVG model requires **language-object pair annotations** for learning 3D-language correlation.
> In SPAZER, the detector is used only to generate candidate object locations; it is not responsible for understanding the language-object correspondence. Instead, all reasoning—from interpreting spatial and semantic cues in the query to selecting the best-matching object—is performed by the VLM in a zero-shot manner, without any 3DVG-specific training.
>
> Thus, SPAZER remains fully consistent with the zero-shot 3DVG setting, in line with previous works such as ZSVG3D (24'CVPR) and SeeGround (25'CVPR), which also adopt pre-trained Mask3D. For fair comparison, we likewise use Mask3D, allowing a direct and equitable comparison of different models’ grounding reasoning capabilities.
>
> ## W2 & Q1: How the proposed framework addresses open-vocabulary tasks.
> We address open-vocabulary tasks mainly by leveraging VLM's ability, while utilizing a 3D detector to provide essential layout and spatial localization cues.
>
> Specifically, we adopt the VLM to **interpret the query, reasons about object identity, and infers spatial relations** through rendered 3D views and 2D camera images. While the detector provides only closed-set object categories, our framework alleviates this limitation in two ways: (1) a matching algorithm is used to associate VLM-inferred target class with semantically similar detected categories; (2) even in the presence of noisy detections or implied objects, our retrieval-augmented matching strategy enables the VLM to directly assess each box based on visual cues without relying on the detector’s label.
>
> Therefore, SPAZER inherits the open-vocabulary capability directly from the VLM: it understands language, observes the scene and selects the best-matching object.
>
> ## W3 & Q2: Text-matching anchor filtering cannot effectively identify implied object targets.
> First, we want to clarify that the process of identifying target object class and anchor filtering is primarily conducted by the VLM. The text matching algorithm only plays an **auxiliary role** in this process. It serves as a lightweight filtering strategy, attempting to match the VLM-inferred target class with the detector-provided class labels to help remove irrelevant objects during visual anchor annotation. In this way, the VLM can focus its reasoning on the most relevant regions of the scene.
>
> Of course, the matching step can be replaced with VLM-based reasoning, and we consider the following two cases:
> 1. **VLM-matching**: Feed all detected object classes to the VLM, and it determines the matched class. It achieves similar results as text-matching. We believe it is because the matching step is relative easy and a simple text-matching algorithm can handle it effectively while being more efficient than using a VLM.
> 2. **w/o matching**: Annotate all anchors (usually > 100) and VLM decides how to filter out irrelevant ones by itself. In this setting, the accuracy drops significantly due to the large amount of redundant information in the raw detection results. Providing the VLM with too many annotated anchors makes it difficult to focus on the key regions.
>
> | Filtering Strategy | Acc@25 (%) | Acc@50 (%) |
> |--------------------|------------|------------|
> | Ours               | **57.2**   | **48.8**   |
> | VLM-matching       | 56.4       | 48.4       |
> | w/o matching        | 38.4       | 32.8       |
>
> So our design is an effective **combination of text-matching and VLM reasoning**: when the detector's predicted class aligns with the VLM-inferred class, we leverage the detector's class prior to filter out irrelevant objects, thereby greatly improving the reasoning efficiency of the VLM. On the other hand, when the detector's predictions are unreliable or the VLM struggles to infer the query object’s category, our RAF module enables the VLM to directly determine the class of each box based on its own visual observations, allowing it to recognize implied object targets.
>
> ***Case study of identifying implied objects***:
>
> **Query Text**:
> "I am thirsty. Please bring me something from my desk."
>
> **Scene**: ScanNet scene_0050_00
>
> > **View Selection**
> > "The query asks for something to drink from the desk. In view 4, the desk is clearly visible, and there is **a bottle on it which could be a drink**. This view provides the clearest view of the desk and the items on it, making it the best choice to identify the target object."
>
> > **Candidate Screening**
> > "The query suggests looking for objects related to drinking, **such as bottles, cups, or glasses, that are located on the desk**. In the images, object 37 is a water bottle, which is the most direct match for quenching thirst. Object 66 appears to be a mug, which is also suitable for drinking. Object 60 and 77 seem to be cups or glasses. Object 1 is on the desk and could potentially be a container, although less clear."
>
> > **Joint Decision-Making**
> > "The query suggests looking for a bottle or cup.  In the global view image, **object 37 is on the desk and appears to be a bottle**, matching the query. Camera views confirm that **object 37 is consistently shown on the desk near object 1 (a laptop)**. Other objects like 66 (printer) and 1 (laptop) do not match the description. Therefore, **object 37 is the best match**."
>
> **GT**: object 37 (a bottle of water on the desk)
>
> We sincerely apologize for not being able to include more intuitive visual illustrations in the current submission. We will include clearer visualizations and examples in the revised version to facilitate better understanding of our framework in handling implied objects.
>
> ## W4 & Q3: Broader evaluations in out-of-scannet scenarios.
> To address this concern, we conducted additional experiments on the RIORefer [1] benchmark, which is based on 3RScan [2] and differs significantly from ScanNet in terms of scene layouts, object types, and scanning conditions.
>
> In this setting, both methods use the same detector trained on ScanNet. Besides, Cross3DVG is trained on ScanRefer using language-object annotations, while our SPAZER requires no 3DVG training.
>
> | Method         | Training Data | Setting     | Acc@25 (%) | Acc@50 (%) |
> |----------------|---------------|-------------|--------------|-------------|
> | Cross3DVG      | ScanRefer     | Supervised  | 29.2         | 14.4        |
> | **SPAZER (Ours)** | -             | Zero-shot   | **34.0**     | **16.4**    |
>
> Here, SPAZER outperforms the supervised baseline even across domains, indicating strong cross-dataset generalization and robustness to distribution shifts. We will include these results and corresponding discussion in the revised manuscript.
>
> We believe SPAZER performs well in cross-dataset scenarios because it relies on VLM's broad visual-language knowledge, rather than dataset-specific labels or training. This allows it to understand diverse object appearances and natural language queries, even in unfamiliar environments.
>
> ## W5: Regarding the supplementary material.
> We sincerely thank the reviewer for pointing this out. The supplementary material was indeed prepared but appears to have been unintentionally omitted during submission. It contains the following sections:
>
> - **A. Implementation Details**
>   - Computational resources
>   - Model configuration and prompt design
>
> - **B. Additional Experimental Results and Analysis**
>   - **B.1**: Error Type Analysis
>     *Spatial relation reasoning remains the major challenge.*
>   - **B.2**: Performance on Subset vs. Full Dataset
>     *Our method and prior works exhibit consistent trends across two dataset partitions.*
>   - **B.3**: Inference Time
>     *SPAZER achieves ~2× faster inference compared to VLM-Grounder.*
>
> - **C. Limitations and Broader Impact**
>   - **C.1**: Failure Case Analysis
>   - **C.2**: Broader Impact Discussion
>
> We appreciate the opportunity to clarify these points and will revise both the main paper and supplementary material to address all reviewer feedback. The complete supplementary will be properly included in the next version.
>
> [1] Miyanishi, Taiki, et al. "Cross3dvg: Cross-dataset 3d visual grounding on different rgb-d scans." 3DV. 2024.
>
> [2] Wald, Johanna, et al. "Rio: 3d object instance re-localization in changing indoor environments." ICCV. 2019.

---

> > ### Author Response · Authors · 2025-08-07
> >
> > Dear Reviewer 8wRM,
> >
> > We sincerely appreciate the time and thoughtful feedback you have provided throughout the review process.
> >
> > As the rebuttal deadline approaches in two days, we would like to kindly check whether our latest response has sufficiently addressed your concerns.
> >
> > Thank you once again for your time and effort.
> >
> > The Authors

---

> > ### Comment · Reviewer_8wRM · 2025-08-07
> >
> > Thanks for the authors' detailed explanation. Most of my concerns have been addressed. I decide to raise my rating to borderline accept. In addtion, I strongly encourage the authors to include these details in the paper.

---

### Decision · Program_Chairs · 2025-09-17

**Decision:**

Accept (poster)

**Comment:**

This paper proposes a new VLM-driven agent SPAZER, by integrating both spatial and semantic reasoning for robust zero-shot grounding without 3D-specific training. Through holistic scene analysis and joint 3D-2D decision-making, SPAZER surpasses prior state-of-the-art methods on benchmark datasets, achieving accuracy improvements.

The main strengths of the work: The work introduces a novel framework that effectively integrates 3D and 2D reasoning for visual grounding, yielding state-of-the-art results on key benchmarks. It is well-motivated with comprehensive experiments.

The detailed rebuttal effectively addresses the reviewers' major concerns, leading to the final decision to accept the work.